# Association of sleep duration in middle and old age with incidence of dementia

Séverine Sabia 1,2✉, Aurore Fayosse 1, Julien Dumurgier1,3, Vincent T. van Hees4, Claire Paquet3, Andrew Sommerlad5,6, Mika Kivimäki 2,7, Aline Dugravot1 & Archana Singh-Manoux 1,2

Sleep dysregulation is a feature of dementia but it remains unclear whether sleep duration prior to old age is associated with dementia incidence. Using data from 7959 participants of the Whitehall II study, we examined the association between sleep duration and incidence of dementia (521 diagnosed cases) using a 25-year follow-up. Here we report higher dementia risk associated with a sleep duration of six hours or less at age 50 and 60, compared with a normal (7 h) sleep duration, although this was imprecisely estimated for sleep duration at age 70 (hazard ratios (HR) 1.22 (95% confidence interval 1.01–1.48), 1.37 (1.10–1.72), and 1.24 (0.98–1.57), respectively). Persistent short sleep duration at age 50, 60, and 70 compared to persistent normal sleep duration was also associated with a 30% increased dementia risk independently of sociodemographic, behavioural, cardiometabolic, and mental health factors. These findings suggest that short sleep duration in midlife is associated with an increased risk of late-onset dementia.

[1] Université de Paris, Inserm U1153, Epidemiology of Ageing and Neurodegenerative diseases, Paris, France. [2] Department of Epidemiology and Public Health, University College London, London, UK. [3] Université de Paris, Inserm U1144, Cognitive Neurology Center, GHU APHP Nord Lariboisière – Fernand Widal Hospital, Paris, France. [4] Accelting, Andorrastraat 13, Almere, The Netherlands. [5] Division of Psychiatry, University College London, London, UK. [6] Camden and Islington NHS Foundation Trust, London, UK. [7] Clinicum, University of Helsinki, Helsinki, Finland. ✉email: severine.sabia@inserm.fr

C changes in sleep patterns are common in persons with Alzheimer's disease and other dementias[1,2]. These changes are believed to result from sleep–wake cycle dysregulation due to pathophysiological processes in dementia, particularly those affecting the hypothalamus and the brainstem[3]. Besides sleep disturbance, there is growing interest in the association between sleep duration and dementia[1,2,4,5]. Observational studies show both short and long sleep duration to be associated with the increased risk of cognitive decline and dementia[1,4]. Some studies also report change in sleep duration in older adults to be associated with the risk of dementia[1,6–8].

Much of the evidence on the association between sleep duration and dementia comes from studies with a follow-up of <10 years. As most dementias are characterized by pathophysiological changes over 20 years or more[9,10], studies with a long follow-up are needed to provide an insight into the association between sleep duration and subsequent dementia. Among studies with a follow-up of 10 years or longer[7,11–15], many are based on participants aged 65 years and older at baseline[7,13–15], not allowing the examination of the importance of sleep characteristics earlier in the lifecourse. The number of dementia cases in the short and long sleep groups in these studies is often small[7,11–13], leading to imprecise associations due to limited statistical power. Whether the patterns of change in sleep duration leading up to old age is associated with incidence of dementia is also unclear. In addition, the role of mental health in the association of sleep duration with dementia merits consideration[16,17] as mental health disorders are associated with both sleep duration[18] and cognitive health[19].

In this work, we use data from the Whitehall II cohort study spanning 30 years to examine the association of sleep duration at age 50, 60, and 70 with incident dementia, and to investigate whether patterns of change in sleep duration over this period were associated with dementia. In our analyses, we examine whether mental disorders in midlife affects the association of sleep duration with dementia. Given potential bias in self-reported measures of sleep duration, we examine the association between objectively assessed sleep duration and risk of dementia in a sub-sample[20] of the study. We find that short sleep duration in midlife is associated with the higher risk of dementia later in life, independently of sociodemographic, behavioural, cardiometabolic, and mental health factors.

## Results

Among the 10,308 persons recruited to the study in 1985–1988, 7959 had data on sleep duration and covariates when they were 50 years (mean age at clinical assessment = 50.6 years, standard deviation (SD) = 2.6 years; flow chart in Fig. 1). Among these participants, 521 developed dementia over a mean follow-up period of 24.6 (SD = 7.0) years, the follow-up being 25.7 (SD = 5.1) years in dementia cases and 24.6 (SD = 7.1) years in non-cases. Figure 1 also shows the construction of the analytic sample for sleep duration at age 60 and 70. Cumulative hazards of dementia as a function of sleep duration at age 50, 60, and 70 are presented in Supplementary Fig. 1, and show that most cases of dementia were diagnosed after the age of 70 years. Mean age at diagnosis was 77.1 (SD = 5.6; range = 53.4–87.6) years.

Characteristics of study participants at age 50 are presented in Table 1. Participants with sleep duration of 7 h per night, labelled as normal sleep duration, were more likely to be men (69.1% compared to 67.1% and 61.5% in short (≤6 h) and long (≥8 h) sleepers, respectively), white (91.5% compared to 89.6 and 86.3%), married (77.9% compared to 72.9% and 75.6%), and to have better cardiometabolic and mental health profile. Characteristics of study participants at age 60 and 70 are shown in Supplementary Table 1. As there was no evidence of systematic

sex differences (interaction of sex with sleep duration variables, all $P > 0.31$), the analyses were conducted combining men and women and adjusted for sex.

**Association of sleep duration with dementia.** Table 2 presents the association of sleep duration at age 50, 60, and 70 with incident dementia over a mean follow-up of 24.6 (SD = 7.0), 14.8 (SD = 5.9), and 7.5 (SD = 4.7) years, respectively. The lowest dementia incidence per 1000 person-years was observed among those who slept 7 h per night, irrespective of the age at which sleep duration was measured. In analysis adjusted for sociodemographic factors, short sleep duration was associated with the higher risk of incident dementia at all ages (all $P < 0.02$). Further adjustment for health behaviours and cardiometabolic and mental health factors attenuated associations, but there remained an association for short sleep at 50 (hazard ratio (HR) = 1.22, 95% confidence interval (CI) = 1.01–1.48) and 60 years (HR = 1.37, 95% CI = 1.10–1.72). There was no clear evidence of an association between long sleep duration and incident dementia.

A total of 6875 participants were alive, free of dementia at age 70, and had at least two out of the three measures of sleep duration at age 50, 60, and 70. Among them, 426 had incident dementia over a mean follow-up period of 7.4 (SD = 4.7) years. Using these data on sleep duration, six trajectories were identified and labelled as: persistent short sleep, persistent normal sleep, persistent long sleep, change from short to normal sleep, change from normal to long sleep, and change from normal to short sleep (Table 3). Persistent short sleep duration was associated with an increased risk of dementia (HR = 1.30, 95% CI = 1.00–1.69) compared to those with persistent normal sleep duration (Table 4). There was also a signal of higher risk in participants with persistent long sleep and those who reported short sleep at least once, but the associations did not reach statistical significance. Analyses restricted to participants without a history of mental illness before age 65 years (Table 5) showed associations of sleep duration and change in sleep duration with subsequent dementia to be similar to that in the main analyses.

Data on accelerometer-assessed sleep duration collected in 2012–2013 were available on 4267 participants. Data were excluded due to absence of sleep log ($n = 140$), technical problem ($n = 175$), significant non-wear ($n = 23$), and missing covariates ($n = 41$) leading to a total of 3888 participants included in the analysis, among whom 111 developed dementia over a mean 6.4-year follow-up period. Characteristics of this analytical sample are presented in Supplementary Table 2. The Pearson's correlation between questionnaire and accelerometer-assessed measures of sleep duration in 2012–2013 was 0.41 ($P < 0.001$). The association between accelerometer-assessed sleep duration, modelled as a continuous variable, and dementia is shown in Fig. 2. Compared to 7 h of sleep, sleep duration <6 h per night was associated with the higher risk of dementia, whereas sleep duration >8 h per night was not associated with risk of dementia (Fig. 2a, Source data), the estimates for long sleep duration were further attenuated when missing data were taken into account (Fig. 2b, Source data). The analysis of these data using tertiles, with the second tertile as the reference (6 h 14 min–7 h per night), showed sleep duration in the first tertile (1 h 16 min–6 h 13 min) to be associated with an increased risk of dementia (HR = 1.63, 95% CI = 1.04–2.57), while there was no association with dementia in the highest tertile (7 h 1 min–10 h 6 min), Table 6.

**Sensitivity analysis.** Participants who could not be included in the analyses due to missing data ($N = 1987$) were older at recruitment to the study in 1985–1988 (45.4 vs 41.5 years, $P < 0.001$), and more likely to develop dementia over the follow-up (6.6% vs 4.0%, $P <$

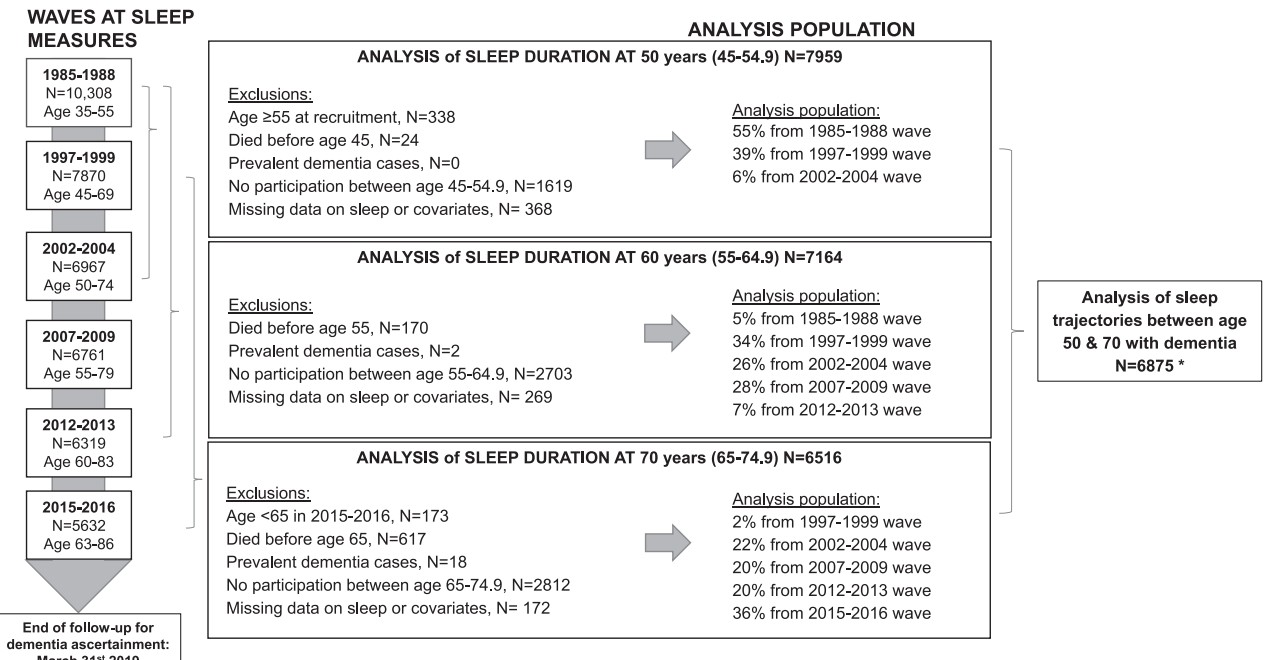

**Fig. 1 Flow chart for population selection.** This figure represents the sample selection for the analysis of sleep duration at age 50, 60, and 70, as well as for the analysis of sleep trajectories.

**Table 1 Characteristics of the study population at age 50.**

| | Sleep duration at age 50 | | | | Dementia status at end of follow-up | | |
| --- | --- | --- | --- | --- | --- | --- | --- |
| | Short: ≤6 h | Normal: 7 h | Long: ≥8 h | P | No dementia | Dementia | P |
| N | 3149 | 3624 | 1186 | | 7438 | 521 | |
| Women | 1035 (32.9) | 1118 (30.9) | 456 (38.5) | <0.001 | 2390 (32.1) | 219 (42.0) | <0.001 |
| Ethnicity, Nmon-white | 326 (10.4) | 309 (8.5) | 163 (13.7) | <0.001 | 719 (9.7) | 79 (15.2) | <0.001 |
| Less than secondary school diploma | 1522 (48.3) | 1722 (47.5) | 575 (48.5) | 0.75 | 3509 (47.2) | 310 (59.5) | <0.001 |
| Married/cohabiting | 2296 (72.9) | 2823 (77.9) | 897 (75.6) | <0.001 | 5647 (75.9) | 369 (70.8) | 0.009 |
| Moderate alcohol drinking (1–14 units per week) | 1695 (53.8) | 2042 (56.4) | 618 (52.1) | 0.02 | 4082 (54.9) | 273 (52.4) | 0.27 |
| Current smokers | 519 (16.5) | 552 (15.2) | 168 (14.2) | 0.13 | 1138 (15.3) | 101 (19.4) | 0.013 |
| Moderate-to-vigorous physical activity (h), M (SD) | 3.3 (3.9) | 3.4 (3.5) | 3.2 (3.6) | 0.14 | 3.3 (3.7) | 3.3 (3.7) | 0.78 |
| Daily fruit and vegetable consumption | 1953 (62.0) | 2341 (64.6) | 777 (65.5) | 0.03 | 4774 (64.2) | 297 (57.0) | 0.001 |
| BMI ≥ 30 kg/m² | 414 (13.2) | 355 (9.8) | 120 (10.1) | <0.001 | 823 (11.1) | 66 (12.7) | 0.26 |
| Diabetes | 97 (3.1) | 80 (2.2) | 22 (1.9) | 0.02 | 182 (2.5) | 17 (3.3) | 0.25 |
| Hypertension | 779 (24.7) | 783 (21.6) | 288 (24.3) | 0.006 | 1700 (22.9) | 150 (28.8) | 0.002 |
| Cardiovascular disease | 105 (3.3) | 91 (2.5) | 28 (2.4) | 0.07 | 214 (2.9) | 10 (1.9) | 0.20 |
| GHQ depression | 577 (18.3) | 439 (12.1) | 143 (12.1) | <0.001 | 1070 (14.4) | 89 (17.1) | 0.09 |
| CNS medications | 169 (5.4) | 130 (3.6) | 59 (5.0) | 0.001 | 322 (4.3) | 36 (6.9) | 0.006 |
| Mental disorders before age 65 | 259 (8.2) | 231 (6.4) | 93 (7.8) | 0.01 | 541 (7.3) | 42 (8.1) | 0.51 |

Values are No. (%) unless stated otherwise. Two-sided Ps for heterogeneity were estimated using $\chi^2$ test for categorical variables, and ANOVA for continuous variables by sleep duration groups and Student's t test by dementia groups for continuous variables.
BMI body mass index, CNS central nervous system, GHQ general health questionnaire, M mean, SD standard deviation.

0.001), although no differences were observed in relation to sex (P = 0.81) or sleep duration (P = 0.65) categories. Analyses using inverse-probability weighting to account for missing data led to results that were consistent with those in the main analysis, as well as the accelerometer sub-study (Table 7).

Adding apolipoprotein E (APOE) ε4 as a covariate in the analysis did not alter observed associations between sleep duration at age 50, 60, and 70 and risk of dementia (Supplementary Table 3). Use of dementia without history of cardiovascular disease as a proxy for Alzheimer's disease type of dementia (N cases = 404, 77.5% of the 521 all cause dementia

cases) yielded results similar to those in the main analysis although the estimates were imprecise due to smaller numbers in analysis (Supplementary Table 4).

## Discussion

This longitudinal study of nearly 8000 participants with repeat data on sleep duration and a long follow-up for dementia suggests short sleep duration in midlife to be associated with the increased risk of incident dementia. This finding was confirmed in analysis using sleep duration measured by an accelerometer. Measurement

**Table 2 Association between sleep duration at age 50, 60, and 70 years and incidence of dementia.**

| | N cases/N total | Incidence rate per 1000 persons-years | Model 1: adjusted for sociodemographic variables[a] | | Model 1 + behavioural factors[b] | | Model 1 + health-related factors[c] | | Fully adjusted model | |
|---|---|---|---|---|---|---|---|---|---|---|
| | | | HR (95%CI) | P value[d] | HR (95%CI) | P value[d] | HR (95%CI) | P value[d] | HR (95%CI) | P value[d] |
| Sleep duration at age 50[e] | 521/7959 | | | | | | | | | |
| Short: ≤6 h | 211/3149 | 2.8 (2.4–3.2) | 1.28 (1.06–1.55) | 0.01 | 1.27 (1.05–1.54) | 0.01 | 1.22 (1.01–1.48) | 0.04 | 1.22 (1.01–1.48) | 0.04 |
| Normal: 7 h | 219/3624 | 2.4 (2.1–2.7) | 1 (ref.) | | 1 (ref.) | | 1 (ref.) | | 1 (ref.) | |
| Long: ≥8 h | 91/1186 | 3.0 (2.4–3.7) | 1.25 (0.98–1.59) | 0.08 | 1.25 (0.98–1.60) | 0.07 | 1.24 (0.97–1.59) | 0.09 | 1.25 (0.98–1.60) | 0.07 |
| Sleep duration at age 60[e] | 409/7164 | | | | | | | | | |
| Short: ≤6 h | 192/2759 | 4.7 (4.0–5.4) | 1.48 (1.19–1.84) | <0.001 | 1.46 (1.17–1.82) | 0.001 | 1.38 (1.11–1.73) | 0.004 | 1.37 (1.10–1.72) | 0.005 |
| Normal: 7 h | 142/2988 | 3.2 (2.7–3.7) | 1 (ref.) | | 1 (ref.) | | 1 (ref.) | | 1 (ref.) | |
| Long: ≥8 h | 75/1417 | 3.6 (2.8–4.4) | 1.15 (0.87–1.52) | 0.33 | 1.17 (0.88–1.55) | 0.28 | 1.13 (0.85–1.50) | 0.39 | 1.15 (0.87–1.52) | 0.34 |
| Sleep duration at age 70[e] | 392/6516 | | | | | | | | | |
| Short: ≤6 h | 171/2429 | 9.3 (7.9–10.7) | 1.33 (1.06–1.68) | 0.004 | 1.29 (1.03–1.63) | 0.005 | 1.26 (1.00–1.60) | 0.04 | 1.24 (0.98–1.57) | 0.10 |
| Normal: 7 h | 131/2578 | 6.8 (5.6–7.9) | 1 (ref.) | | 1 (ref.) | | 1 (ref.) | | 1 (ref.) | |
| Long: ≥8 h | 90/1509 | 8.1 (6.4–9.7) | 1.22 (0.94–1.60) | 0.39 | 1.13 (0.91–1.55) | 0.34 | 1.18 (0.90–1.55) | 0.22 | 1.15 (0.88–1.51) | 0.60 |

CI confidence intervals, HR hazard ratio, SD standard deviation.
[a]HR estimated from a Cox regression adjusted for age (timescale), sex, ethnicity, education, and marital status.
[b]Additionally adjusted for alcohol consumption, physical activity, smoking status, and fruit and vegetable consumption.
[c]Additionally adjusted for BMI, hypertension, diabetes, cardiovascular disease, GHQ depression, and CNS medications.
[d]Two-sided P value for HR in comparison with the reference (ref.) category, without adjustment for multiple comparisons.
[e]Follow-up: at age 50, mean = 24.6 years, SD = 7.0 years, range = 0.18–33.6 years; at age 60, mean = 14.8 years, SD = 5.9 years, range = 0.11–33.6 years; at age 70, mean = 7.5 years, SD = 4.7 years, range = 0.04–21.8 years.

of sleep duration at age 50, 60, and 70 years along with change in sleep duration over this period provides consistent results for increased risk of dementia in those with short sleep. A further key finding is that the association between short sleep duration and dementia is not attributable to mental health.

Two recent meta-analyses suggest a U-shaped association between sleep duration and incident dementia, with lower risk in people sleeping 7 h per night, and greater risk among those with shorter sleep and also among those with longer sleep duration[4,5]. The studies included in these meta-analyses had a follow-up duration ranging from 3 to 23 years, but most studies had a follow-up of <10 years[4,5]. Dementia at older ages is characterized by a long preclinical period[9,20], and studies with a follow-up of 10 years are likely to be subject to reverse causation bias, whereby the putative risk factor is affected by the disease process in dementia.

Some studies[11–15] are noteworthy because of their long follow-up. In the Kuopio Ischaemic Heart Disease study on 2386 men aged 42–62 years at assessment of sleep duration, there was no association between sleep duration (categorized as ≤6.5, 7–8, and ≥8.5 h) and incidence of dementia over a 22-year mean follow-up[12]. On the other hand, a U-shaped association between sleep duration and dementia was observed in three studies, where the mean age at sleep duration assessment varied between 50 and 73 years, and follow-up between 15 and 23 years[11,13,14]. In the Framingham Heart study, long (N = 96) but not short (N = 209) sleep duration at mean age 72 years was associated with dementia over a 10-year follow-up[7]. In a large-scale Swedish cohort of 28,775 individuals aged 65 years and older, the association between long sleep duration and dementia over a 13-year follow-up was completely attenuated after cases occurring in the first 5 years of follow-up were excluded from the analysis, highlighting the role of reverse causation bias[15]. All these studies use a wide age range at baseline for assessment of sleep duration and it is likely that the older participants, those most likely to be diagnosed with dementia over the follow-up, are already subject to preclinical changes that characterize dementia. This long preclinical period of dementia makes it less amenable to research using gold-standard approaches, such as randomized controlled trials. Observation studies are not ideally suited for causal inference, but careful analysis that considers time between exposure and outcome can be helpful in determining changes over time in the nature of the association between the exposure of interest and dementia.

In the present study, we used an innovative approach consisting of extracting data on sleep duration at age 50, 60, and 70 in order to remove the uncertainty in estimations caused by inclusion of a wide age range at the start of the follow-up. The pattern of associations in relation to short sleep duration and dementia was similar at age 50, 60, and 70, even though sleep duration at age 70 was not associated with dementia after adjustment for health-related factors. Analysis of trajectories of sleep duration using data from sleep duration at age 50, 60, and 70 showed persistent short sleep duration to be associated with an increased risk of dementia. Depression and mood disorders in general are related to changes in sleep and thought to play an important part in the association of sleep duration with dementia[8]. In the present study, adjustment for depressive symptoms and central nervous system (CNS) drugs, as well as analysis undertaken among those without a history of mental disorders did not show mental health to explain the association. Our results are consistent with a recent Mendelian randomization study, where sleep-related phenotypes were found not to be related to mental disorder[17]. Results in our study were robust to adjustments for a wide range of covariates, including cardiovascular disease[16], suggesting a consistent association between short sleep duration in midlife and incident dementia.

**Table 3 Description of sleep duration at age 50, 60, and 70 by groups of trajectories of sleep duration.**

| | N | Sleep duration, mean (standard deviation) (h) | | |
|---|---|---|---|---|
| | | Age 50 | Age 60 | Age 70 |
| Persistent short | 1358 | 6.0 (0.0) | 6.0 (0.0) | 6.0 (0.0) |
| Persistent normal | 2520 | 7.2 (0.4) | 7.0 (0.5) | 6.8 (0.5) |
| Persistent long | 461 | 8.0 (0.0) | 8.0 (0.0) | 7.8 (0.4) |
| Change from short to normal | 1086 | 6.0 (0.0) | 6.7 (0.6) | 6.9 (0.6) |
| Change from normal to long | 946 | 6.9 (0.5) | 7.5 (0.6) | 8.0 (0.6) |
| Change from normal to short | 504 | 7.1 (0.3) | 6.0 (0.1) | 6.0 (0.0) |

There is evidence of a bidirectional association between sleep dysfunction and pathophysiological changes in dementia[21,22], highlighting the need for a longer time frame in studies. The results for short sleep duration seen in our study is likely to involve several processes[1], including neuroinflammation[23], atherosclerosis[24], alpha-synucleinopathies (dementia with Lewy bodies and Parkinson disease dementia)[25], and impaired amyloid-β clearance[26], possibly due to impaired glymphatic function[27]. Experimental studies support a detrimental effect of sleep deprivation on cognitive performance[28] and β-amyloid (Aβ) clearance[29–31]. Amyloid plaque build-up contributes to poor sleep in older adults through its direct impact on sleep–wake regulator brain regions[21,22]. There is also some evidence of an association of Aβ accumulation with disruption of the circadian rhythm and sleep pattern in cognitively normal adults[32]. Much of the research on mechanisms focusses on sleep disturbance rather than sleep duration, particularly in relation to the years leading up to dementia diagnosis. The mechanisms linking short sleep duration to dementia may be similar[1,33], but a better understanding of how sleep features (duration, disturbance, sleep apnoea, and sleep–wake regulation) over the adult lifecourse shape risk of dementia at older ages is required to identify windows of opportunity for therapeutic interventions to reduce the risk or delay the progression of dementia and its subtypes.

In contrast to some previous studies[7,11,14,15], we did not find strong evidence to support the hypothesis that long sleep duration is associated with dementia. As in most previous studies[7,11–13], the number of long sleepers in our study was small and did not allow a robust estimation of the association with long sleep duration. Two large-scale studies based on individuals with a mean age of 72 years, followed for 12 and 14 years reported sleep duration >9 h to be associated with the increased risk of dementia[14,15]. Given the age of these individuals at the start of follow-up, more data on long sleepers in midlife and old age are needed to draw conclusions on the importance of long sleep duration for dementia.

Strengths of the present study include repeat measures of sleep duration and a long follow-up for dementia allowing the examination of age at assessment of sleep duration, and change therein to provide insight into the nature of the association between sleep duration and dementia. Use of an objective measure of sleep duration confirmed the main findings that were based on self-reported sleep duration. We also undertook sensitivity analysis using inverse-probability weighting to take missing data into account, and results from these analyses were in accordance with the main findings.

Assessment of dementia cases via electronic health records is not ideal but has some advantages. It allows ascertainment of dementia status in all participants and not only those who agree to continue to participate in multiple waves of data collection over time. However, this method may misclassify some dementia cases, particularly milder cases of dementia[34], although the misclassification

**Table 4 Association of of trajectories of sleep duration (using data on sleep duration at 50, 60, and 70 years, N cases/N total = 426/6875) with incidence of dementia.**

| Trajectories of sleep duration between age 50 and 70[a] | N cases/ N total | Incidence rate per 1000 persons-years | Model 1: adjusted for sociodemographic variables[b] | | Model 1 + behavioural factors[c] | | Model 1 + health-related factors[d] | | Fully adjusted model | |
|---|---|---|---|---|---|---|---|---|---|---|
| | | | HR (95%CI) | P value[e] | HR (95%CI) | P value[e] | HR (95%CI) | P value[e] | HR (95%CI) | P value[e] |
| Persistent short | 103/1358 | 10.5 (8.5-12.5) | 1.40 (1.08-1.81) | 0.01 | 1.35 (1.05-1.75) | 0.02 | 1.32 (1.02-1.72) | 0.03 | 1.30 (1.00-1.69) | 0.048 |
| Persistent normal | 141/2520 | 7.3 (6.1-8.5) | 1 (ref.) | | 1 (ref.) | | 1 (ref.) | | 1 (ref.) | |
| Persistent long | 35/461 | 9.9 (6.6-13.1) | 1.32 (0.91-1.91) | 0.15 | 1.27 (0.88-1.85) | 0.20 | 1.32 (0.91-1.91) | 0.15 | 1.28 (0.88-1.85) | 0.20 |
| Change from short to normal | 61/1086 | 8.2 (6.1-10.2) | 1.23 (0.91-1.66) | 0.18 | 1.21 (0.90-1.64) | 0.21 | 1.21 (0.90-1.64) | 0.21 | 1.20 (0.89-1.63) | 0.23 |
| Change from normal to long | 47/946 | 7.1 (5.0-9.1) | 1.04 (0.75-1.45) | 0.81 | 1.03 (0.74-1.44) | 0.85 | 1.03 (0.74-1.44) | 0.86 | 1.02 (0.73-1.42) | 0.90 |
| Change from normal to short | 39/504 | 9.6 (6.6-12.6) | 1.21 (0.84-1.73) | 0.30 | 1.17 (0.82-1.68) | 0.38 | 1.15 (0.80-1.65) | 0.44 | 1.13 (0.79-1.62) | 0.50 |

CI confidence intervals, HR hazard ratio, SD standard deviation.
[a]Follow-up: mean = 7.4 years, SD = 4.7 years, range = 0.1-21.8 years.
[b]HR estimated from a Cox regression adjusted for age (timescale), sex, ethnicity, education, and marital status.
[c]Additionally adjusted for alcohol consumption, physical activity, smoking status, and fruit and vegetable consumption.
[d]Additionally adjusted for BMI, hypertension, diabetes, cardiovascular disease, GHQ depression, and CNS medications.
[e]Two-sided P value for HR in comparison with the reference (ref.) category, without adjustment for multiple comparisons.

**Table 5 Association between sleep duration and dementia: analysis restricted to participants without mental disorders before 65 years.**

| | Total population | | | Among those free of mental disorders before 65 years | | |
|---|---|---|---|---|---|---|
| | N cases/N total | HR[a] (95% CI) | P value[b] | N cases/N total | HR[a] (95% CI) | P value[b] |
| Sleep duration at age 50 | | | | | | |
| Short: ≤6 h | 211/3149 | 1.22 (1.01-1.48) | 0.04 | 195/2890 | 1.25 (1.03-1.53) | 0.03 |
| Normal: 7 h | 219/3624 | 1 (ref.) | | 202/3393 | 1 (ref.) | |
| Long: ≥8 h | 91/1186 | 1.25 (0.98-1.60) | 0.07 | 82/1093 | 1.23 (0.95-1.60) | 0.11 |
| Sleep duration at age 60 | | | | | | |
| Short: ≤6 h | 192/2759 | 1.37 (1.10-1.72) | 0.005 | 170/2520 | 1.28 (1.02-1.62) | 0.04 |
| Normal: 7 h | 142/2988 | 1 (ref.) | | 134/2779 | 1 (ref.) | |
| Long: ≥8 h | 75/1417 | 1.15 (0.87-1.52) | 0.34 | 72/1288 | 1.20 (0.90-1.60) | 0.22 |
| Sleep duration at age 70 | | | | | | |
| Short: ≤6 h | 171/2429 | 1.24 (0.98-1.57) | 0.10 | 159/2237 | 1.28 (1.00-1.63) | 0.049 |
| Normal: 7 h | 131/2578 | 1 (ref.) | | 122/2418 | 1 (ref.) | |
| Long: ≥8 h | 90/1509 | 1.15 (0.88-1.51) | 0.60 | 85/1391 | 1.22 (0.93-1.62) | 0.16 |
| Trajectories of sleep duration between age 50 and 70 | | | | | | |
| Persistent short | 103/1358 | 1.30 (1.00-1.69) | 0.048 | 94/1253 | 1.29 (0.98-1.69) | 0.06 |
| Persistent normal | 141/2520 | 1 (ref.) | | 132/2353 | 1 (ref.) | |
| Persistent long | 35/461 | 1.28 (0.88-1.85) | 0.20 | 32/430 | 1.26 (0.85-1.85) | 0.25 |
| Change from short to normal | 61/1086 | 1.20 (0.89-1.63) | 0.23 | 59/1006 | 1.28 (0.94-1.74) | 0.12 |
| Change from normal to long | 47/946 | 1.02 (0.73-1.42) | 0.90 | 45/870 | 1.11 (0.79-1.56) | 0.55 |
| Change from normal to short | 39/504 | 1.13 (0.79-1.62) | 0.50 | 36/462 | 1.16 (0.80-1.56) | 0.43 |

*CI* confidence intervals, *HR* hazard ratio, *SD* standard deviation.
[a]HR estimated from a Cox regression adjusted for age (timescale), sex, ethnicity, education, and marital status, alcohol consumption, physical activity, smoking status, and fruit and vegetable consumption, BMI, hypertension, diabetes, cardiovascular disease, GHQ depression, and CNS medications.
[b]Two-sided *P* value for HR in comparison with the reference (ref.) category, without adjustment for multiple comparisons.

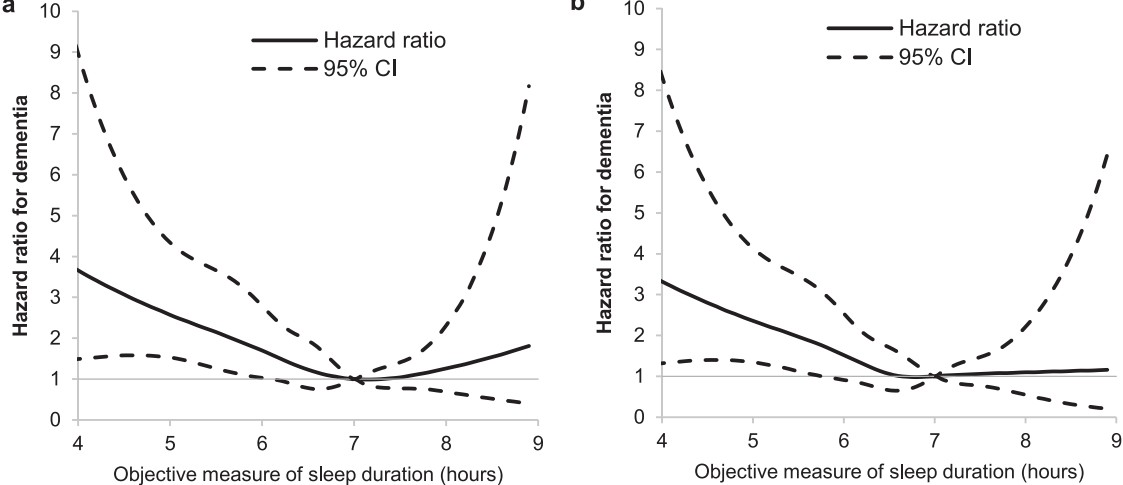

**Fig. 2 Association of objectively assessed sleep duration (2012–2013, N cases/N total = 111/3888) with incident dementia over a mean follow-up of 6.4 (SD = 1.0) years: accelerometer sub-study. a** The hazard ratio for dementia (black plain line) with the corresponding 95% confidence interval (black dotted line) as a function of sleep duration from a Cox regression adjusted for age (timescale), sex, ethnicity, education, marital status, alcohol consumption, physical activity, smoking status, fruit and vegetable consumption, BMI, hypertension, diabetes, cardiovascular disease, GHQ depression, and CNS medications (Source data). **b** The hazard ratio for dementia (black plain line) with the corresponding 95% confidence interval (black dotted line) as a function of sleep duration from a Cox regression using inverse-probability weighting to account for missing data and adjusted for age (timescale), sex, ethnicity, education, marital status, alcohol consumption, physical activity, smoking status, fruit and vegetable consumption, BMI, hypertension, diabetes, cardiovascular disease, GHQ depression, and CNS medications (Source data).

is likely to be independent from measures of sleep duration. Data on dementia type were incomplete in the records and our use of a proxy measure of Alzheimer's disease dementia may not be precise despite the proportion of cases being in accordance with that in the general population[10]. The observational nature of the study cannot preclude residual confounding despite our adjustment for a large set of covariates. A further limitation is that Whitehall II study participants were all in employment at recruitment and are

healthier than the general population, both in terms of risk factor profiles and incidence of disease. However, the association between risk factors and disease of interest is not necessarily affected[35]. A previous study showed that the association between cardiovascular risk factors and risk of CVD in the Whitehall II study was similar to that in general population studies[36].

There is a widely acknowledged association between sleep and cognitive function[1], primarily due to the role of sleep in learning

**Table 6 Association of tertiles of objectively assessed sleep duration with dementia over a mean follow-up of 6.4 (SD = 1.0; range = 0.1-7.4) years: accelerometer sub-study (N cases/N total = 111/3888).**

| | N cases/N total | Incidence rate per 1000 persons-years | Model 1: adjusted for sociodemographic variables[a] HR (95%CI) | P value[d] | Model 1 + behavioural factors[b] HR (95%CI) | P value[d] | Model 1 + health-related factors[c] HR (95%CI) | P value[d] | Fully adjusted model HR (95%CI) | P value[d] |
|---|---|---|---|---|---|---|---|---|---|---|
| Tertile 1: 1 h 16 min–6 h 13 min | 53/1296 | 6.5 (4.7–8.2) | 1.68 (1.08–2.63) | 0.02 | 1.71 (1.09–2.67) | 0.02 | 1.61 (1.03–2.53) | 0.04 | 1.63 (1.04–2.57) | 0.03 |
| Tertile 2: 6 h 14 min–7 h 0 min | 31/1296 | 3.8 (2.4–5.1) | 1 (ref.) | | 1 (ref.) | | 1 (ref.) | | 1 (ref.) | |
| Tertile 3: 7 h 1 min–10 h 6 min | 27/1296 | 3.3 (2.0–4.5) | 0.87 (0.52–1.46) | 0.60 | 0.87 (0.52–1.46) | 0.61 | 0.78 (0.46–1.31) | 0.35 | 0.78 (0.46–1.32) | 0.36 |

CI confidence intervals, HR hazard ratio, SD standard deviation, h hour, min minute.
[a] HR estimated from a Cox regression adjusted for age (timescale), sex, ethnicity, education, and marital status.
[b] Additionally adjusted for alcohol consumption, physical activity, smoking status, and fruit and vegetable consumption.
[c] Additionally adjusted for BMI, hypertension, diabetes, cardiovascular disease, GHQ depression, and CNS medications.
[d] Two-sided P value for HR in comparison with the reference (ref.) category, without adjustment for multiple comparisons.

and memory, synaptic plasticity, and waste clearance from the brain[1,37,38]. Whether sleep parameters also affect late-life dementia remains the subject of debate. While incipient dementia is known to affect sleep–wake cycles[1,2,7], the extent to which sleep duration over the adult lifecourse is associated with late-onset dementia is unclear because most studies have not explicitly considered age at assessment of sleep duration or the length of follow-up. Our approach pays attention to both these aspects along with inclusion of a wide array of covariates to show that short sleep duration in midlife is associated with an increased risk of dementia. Public health messages to encourage good sleep hygiene[39] may be particularly important for people at a higher risk of dementia.

## Methods

**Study population.** The Whitehall II study is a cohort study that was established among 10,308 British civil servants (33.1% women, age range 35–55) in 1985–1988 (ref. [40]). Since baseline, follow-up clinical data collection waves have taken place every 4–5 years with each wave lasting ~2 years, with the last wave conducted in 2015–2016. In addition to clinical examinations in the study, data over the follow-up are obtained via linkage to electronic health records of the UK National Health Service (NHS) for participants recruited to the study. The NHS provides most of the health care in the country, including in- and out-patient care, and record linkage is undertaken using a unique NHS identifier held by all UK residents. Data from linked records were updated on an annual basis, until 31st March 2019. Written, informed consent from participants was obtained at each contact. Research ethics approvals were renewed at each wave; the most recent approval was obtained from the University College London Hospital Committee on the Ethics of Human Research (reference number 85/0938).

## Measures

*Sleep duration.* Sleep duration was assessed by questionnaire in 1985–1988, 1997–1999, 2002–2004, 2007–2009, 2012–2013, and 2015–2016 using the question 'how many hours of sleep do you have on an average weeknight?'. Response categories were 5 h or less, 6, 7, 8, and 9 h or more. We extracted sleep duration at age 50, 60, and 70 for each participant across the data waves, allowing a ±5-year margin for each age category. In order to allow sufficient number of dementia cases in each sleep category, we pooled categories of sleep duration as follows: short (≤6 h per night), normal (7 h per night), and long (≥8 h per night)[41].

Trajectories of change in sleep duration between age 50 and 70 were defined using group-based trajectory modelling using the traj-command in Stata[42]. Groups were chosen according to model fit statistics (Bayesian Information Criterion values and average posterior probabilities) and meaningful interpretation[43]. The sleep duration categories in the construction of trajectories were the same as in the main analyses, and were based on participants who were alive and free of dementia at age 70 with at least two out of three measures of sleep duration at age 50, 60, and 70.

The accelerometer substudy was undertaken in 2012–2013 (ref. [44]) on study participants who either attended the central London research clinic or were seen at home in those living in the South-Eastern regions of England. These participants were asked to wear a wrist-worn accelerometer, the GENEActiv (Activinsights Ltd, Kimbolton, UK), during nine consecutive days over 24 h. Sleep duration was estimated using a validated algorithm guided by a sleep log and implemented in R (version 3.6.3) package GGIR version 2.0-1 (ref. [45]); data from the first and last nights were removed leading to data over 7 nights[45]. Usual daily sleep duration was estimated as the mean of sleep duration over the 7 nights and for those with <7 days of measurement, weighted average of sleep duration was calculated according to week days and weekend days[46]. Accelerometer-assessed sleep duration was categorized into tertiles rather than categories used for self-reported sleep as only 167 (six dementia cases) participants had accelerometer-assessed sleep duration of 8 h or more. Study participants were between 60 and 83 years in this sub-study, a one-off addition to the main data collection. Therefore, age-specific analysis as in the self-reported measure of sleep duration was not possible with the accelerometer data.

*Dementia.* Dementia cases were ascertained by linkage to three national registers (the national hospital episode statistics (HES) database, the Mental Health Services Data Set, and the mortality register) up to the 31st of March 2019 using the unique National Health Service (NHS) identification number. Dementia cases were identified based on ICD-10 codes F00-F03, F05.1, G30, and G31. The NHS provides most of the health care, including out- and in-patient care. The sensitivity and specificity of dementia in the NHS HES data is 78.0 and 92.0% (ref. [34]). The sensitivity in our study is likely to be higher as we also used data from the Mental Health Services Data Set, a national database that contains information on dementia for persons in contact with mental health services in hospitals, out-patient clinics, and the community[47]. The first record of dementia diagnosis in any of the three registers was used as date of dementia in the analysis.

**Table 7 Sensitivity analysis: use of inverse-probability weighting (IPW) to account for missing data.**

| | N cases/N total | Main analysis | | Analysis using IPW to account for missing data | |
|---|---|---|---|---|---|
| | | HR[a] (95% CI) | P value[b] | HR[a] (95% CI) | P value[b] |
| Sleep duration at age 50 | | | | | |
| Short: ≤6 h | 211/3149 | 1.22 (1.01–1.48) | 0.04 | 1.21 (0.99–1.48) | 0.06 |
| Normal: 7 h | 219/3624 | 1 (ref.) | | 1 (ref.) | |
| Long: ≥8 h | 91/1186 | 1.25 (0.98–1.60) | 0.07 | 1.28 (0.99–1.66) | 0.06 |
| Sleep duration at age 60 | | | | | |
| Short: ≤6 h | 192/2759 | 1.37 (1.10–1.72) | 0.005 | 1.31 (1.03–1.66) | 0.03 |
| Normal: 7 h | 142/2988 | 1 (ref.) | | 1 (ref.) | |
| Long: ≥8 h | 75/1417 | 1.15 (0.87–1.52) | 0.34 | 1.25 (0.93–1.66) | 0.14 |
| Sleep duration at age 70 | | | | | |
| Short: ≤6 h | 171/2429 | 1.24 (0.98–1.57) | 0.10 | 1.14 (0.88–1.46) | 0.32 |
| Normal: 7 h | 131/2578 | 1 (ref.) | | 1 (ref.) | |
| Long: ≥8 h | 90/1509 | 1.15 (0.88–1.51) | 0.60 | 1.14 (0.85–1.52) | 0.39 |
| Trajectories of sleep duration between age 50 and 70 | | | | | |
| Persistent short | 103/1358 | 1.30 (1.00–1.69) | 0.048 | 1.24 (0.94–1.64) | 0.13 |
| Persistent normal | 141/2520 | 1 (ref.) | | 1 (ref.) | |
| Persistent long | 35/461 | 1.28 (0.88–1.85) | 0.20 | 1.33 (0.90–1.96) | 0.15 |
| Change from short to normal | 61/1086 | 1.20 (0.89–1.63) | 0.23 | 1.21 (0.88–1.69) | 0.24 |
| Change from normal to long | 47/946 | 1.02 (0.73–1.42) | 0.90 | 1.02 (0.71–1.45) | 0.93 |
| Change from normal to short | 39/504 | 1.13 (0.79–1.62) | 0.50 | 1.12 (0.75–1.68) | 0.57 |
| Accelerometer-assessed sleep duration | | | | | |
| Tertile 1: 1 h 16 min–6 h 13 min | 53/1296 | 1.63 (1.04–2.57) | 0.03 | 1.63 (1.03–2.59) | 0.04 |
| Tertile 2: 6 h 14 min–7 h 0 min | 31/1296 | 1 (ref.) | | 1 (ref.) | |
| Tertile 3: 7 h 1 min–10 h 6 min | 27/1296 | 0.78 (0.46–1.32) | 0.36 | 0.82 (0.47–1.41) | 0.82 |

CI confidence intervals, HR hazard ratio, IPW inverse-probability weighting, h hour, min minute.
[a]HR estimated from a Cox regression adjusted for age (timescale), sex, ethnicity, education, and marital status, alcohol consumption, physical activity, smoking status, and fruit and vegetable consumption, BMI, hypertension, diabetes, cardiovascular disease, GHQ depression, and CNS medications.
[b]Two-sided for HR in comparison with the reference (ref.) category, without adjustment for multiple comparisons.

*Covariates.* Sociodemographic variables included age, sex, ethnicity (white and non-white), education (lower primary school or less, lower secondary school, higher secondary school diploma, and university), and marital status (married or cohabiting, and other). Health behaviours included cigarette smoking status (never smoker, ex-smoker, and current smoker), alcohol consumption in the previous week (no alcohol in the previous week, 1–14 units per week, and >14 units per week), time spent in moderate and vigorous physical activity, and frequency of fruits and vegetables consumption (less than daily, once a day, and twice or more a day).

Health-related variables included measures of cardiometabolic risk factors and mental health. Cardiometabolic factors were hypertension (systolic ≥140 or diastolic ≥90 mmHg or use of antihypertensive medication), diabetes mellitus (determined by fasting glucose ≥7.0 mmol/L, reported doctor-diagnosed diabetes, use of diabetes medication, or hospital record), body mass index (BMI, categorized as <20, 20–24.9, 25–29.9, and ≥30 kg/m²) based on height and weight assessment at the clinical examination using standard clinical protocols, and cardiovascular disease (including coronary heart disease and stroke identified using linkage to national hospital records). Mental health factors included current depressive symptomatology defined by the four-item depression subscale of the General Health Questionnaire[48], and self-reported use of CNS medication (anti-depressant, antipsychotic, hypnotic, anxiolytic, or Parkinson medications).

History of mental disorders before age 65 was assessed based on self-reported use of anti-depressants or linkage to national hospital records and mental health registry based on ICD codes F06, F07, F09, F20–48, and F60–69 (excluding F65 and F66).

**Statistical analysis.** Four sets of analyses were undertaken, described below (Fig. 1 for flow chart). All analyses were performed using Cox regression with age as the timescale to model the associations with incident dementia. Participants were included into the analysis if they had data on sleep duration and covariates assessed at the age of interest allowing a ±5-year margin and were free of dementia at this date. Data were censored at date of record of dementia, death, or March 31st 2019, whichever came first. The proportional hazard assumption was verified using Schoenfeld residuals. Analyses were first adjusted for sociodemographic factors, then additionally for health behaviours, and finally for health-related factors.

We first examined the association between sleep duration at age 50, 60, and 70, in separate models, and incident dementia. For these analyses, age of entry was the age at clinical assessment closest to the age of interest from which the sleep duration measure and covariates were drawn.

We examined the association of trajectories of sleep duration, to reflect changes between the age of 50 and 70, with incident dementia with age of entry and covariates drawn from when participants were 70 years.

In order to assess the role of mental disorders in the association of sleep duration with dementia, we repeated the previous analyses, excluding participants with mental disorders before the age of 65. Mental disorders diagnosed after age 65 were not excluded as they could be features of the preclinical period of dementia[49].

The final analyses examined the association between objectively assessed sleep duration and incident dementia. Participants were followed from the age at accelerometer assessment, and covariates were drawn from the same wave of data collection. To examine the shape of the association between accelerometer-assessed sleep duration, measured in rich detail, and incidence of dementia we used restricted cubic spline regressions with Harrell knots[50], Stata command xblc[51], with 7 h as the reference. Then, in further analyses tertiles of accelerometer-assessed sleep duration were used.

We undertook a series of additional/sensitivity analysis. One, we repeated the main analyses using inverse-probability weighting to account for missing data[52]. This involved first calculating the probability of being included in the analytical sample using logistic regression that included demographic, socioeconomic, behavioural factors, as well as sleep duration at recruitment, morbidities including dementia and mortality over follow-up, and stepwise-selected interactions between covariates. The inverse of these probabilities were used as weights in Cox regression. Two, in a sub-sample of participants with data on APOE genotype, we repeated the main analyses with APOE ε4 (1 or 2, vs 0) as a covariate in the analysis. Three, in order to examine whether observed results apply to dementia due Alzheimer's disease, we repeated the analyses using a proxy definition of this outcome. This consisted of defining Alzheimer's disease dementia as dementia cases free of cardiovascular disease (stroke or myocardial infarction)[53] over the follow-up. Analyses 2 and 3 were exploratory and due to smaller numbers in analysis the direction of results rather than statistical significance should be given more importance. All analyses were performed using STATA version 16.1 (StataCorp). A two-sided P value ≤ 0.05 was considered statistically significant. STROBE statement is available in the online Supplement.

**Reporting summary.** Further information on research design is available in the Nature Research Reporting Summary linked to this article.

## Data availability

Data used for analyses comprise data assessed at study clinical examinations in 1985–1988, 1997–1999, 2002–2004, 2007–2009, 2012–2013, and 2015–2016. In addition to clinical examinations in the study, data over the follow-up were obtained via linkage to electronic health records of the UK National Health services for participants recruited to

the study, including hospital episode statistics databased, Mental Health Services Data Set, and the mortality register. Data from linked records were updated on an annual basis, until 31st March 2019. Following NHS Digital guidelines, these data are available for sharing with the scientific community either through the study specific data sharing https://www.ucl.ac.uk/epidemiology-health-care/research/epidemiology-and-public-health/research/whitehall-ii/data-sharing or using the Dementias platform UK https://www.dementiasplatform.uk/for-researchers/data-portal-getting-started-with-cohort-data. Source data are provided with this paper.

## Code availability

Code for statistical analysis is provided in https://doi.org/10.5281/zenodo.4572438.

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

## Acknowledgements

The Whitehall II study is supported by grants from the National Institute on Aging, NIH (R01AG056477, RF1AG062553); the Wellcome Trust (221854/Z/20/Z); the UK Medical Research Council (R024227, S011676); and the British Heart Foundation (RG/16/11/32334). In addition, M.K. was supported by grants from NordForsk (70521, the Nordic Research Programme on Health and Welfare), the Academy of Finland (311492, 329202), and Helsinki Institute of Life Science (H970). A.S. is funded by the UCL/Wellcome Trust Institutional Strategic Support Fund (204841/Z/16/Z) and by the University College London Hospitals' (UCLH) National Institute for Health Research

(NIHR) Biomedical Research Centre (BRC). S.S. is supported by the French National Research Agency (ANR-19-CE36-0004-01).

## Author contributions

S.S. and A.S.-M. developed the hypothesis and study design. S.S., A.F. and A.D. performed statistical analysis. S.S. wrote first and successive drafts of the manuscript. S.S., A. F., J.D., V.T.v.H., C.P., A.S., M.K., A.D. and A.S.-M. contributed to interpretation of data, and critical revision of the manuscript for important intellectual content. A.S.-M. and M. K. obtained funding.

## Competing interests

The authors declare no competing interests.
