## [Peer Review File · Nature Communications]

REVIEWER COMMENTS

Reviewer #1 (Remarks to the Author):

I enjoyed reading the ms.

Table 2: In the age 70 wave, you had 9% fewer subjects compared to the age 60 wave and 18% fewer subjects compared to the age 50 wave, respectively. With this in mind, I would mention in the ms that the lack of significance for the association between self-reported sleep duration and dementia might have been a power issue.

Others have shown that sleep disturbances, such as difficulty initiating or maintaining sleep and sleep apnea are associated with an increased risk for all-cause dementia/AD (PMID 25438949; 21828324). These conditions can result in shorter sleep duration and may account for the observed association between sleep duration and dementia. It would help if you mentioned this.

If I understood your analysis correctly, you have, for instance, investigated the association of sleep duration reported at the age 50 wave with dementia risk during the entire observational period. For me, it is surprising that you did not consider sleep duration as a time-varying exposure in your main analysis? I am pretty sure that you will still see the association between sleep duration and dementia if you run a time-updated cox regression. It is a strength to use time-varying data for exposure and covariates in your analysis!

Regarding the sleep duration stability analysis: Do you see any differences in the dementia risk between those who have stable sleep duration across all waves (i.e., 50, 60, and 70) and those who do not have stable sleep duration?

Some experimental studies have investigated the effects of short periods of sleep loss on brain health. It would help if you discussed these study findings, as they are in line with your findings that insufficient sleep duration harms the brain. (PMID 31915189; 29632177; 24887018).

Given the observed associations between sleep duration and all-cause dementia, I wonder whether you see similar sleep duration and dementia subtypes associations. In this context, AD may be particularly interesting, as accumulating evidence suggests that sleep disruptions (including short sleep duration) increase the risk of AD (e.g., see 26996255).

Table 1: Please expand this table by showing the descriptive for the sleep duration groups at age 60 and 70.

Reviewer #2 (Remarks to the Author):

Decision: major revision

The authors studied a considerable population with the long-term monitor on sleep duration and dementia development and constituted three important findings despite they only concentrated on sleep duration but no other sleep characteristics (such as sleep disorders and sleep quality). The analyses from this cohort strengthened the role of sleep hygiene as one modifiable factor in understanding mechanisms and establishments of preventable strategies of incipient dementias. But some major questions are required to be addressed by the authors before considering accepting.

1. Although only three aspects resulted in censored data, listing max and min follow-up duration in total participants and those at different age-scale groups as well as numbers of cases and controls in different follow-up durations would be good to assess analytic rationality.

2. "Participants with normal sleep duration more likely to have a better cardiovascular and mental health profile" (in the Results), authors needed to make the definitions of cardiovascular and mental health profile in the Methods because they only mentioned categories of health-related factors.

Besides, whether p values were derived from multiple comparisons of group differences in short, normal, and long sleep durations in Table 1 was not stated.

3. How to understand the observations shown to be a non-significant association of ≤ 5 hours sleep duration in participants at age 50 but significant associations in those at age 60 and 70?

What's more, the authors found the decrease or increase in sleep duration between age 60 and 70 were both associated with risk of dementia when merged subgroups, while in e-Table 2, significant estimates were not noted for most increased or decreased durations. I suggest the authors discuss these points.

4. In main analyses, the authors noted that changes in sleep durations between age 60 and 70 were associated with the risk of dementia without stating details for those who had changed durations. Some participants might have persistent short durations between age 50 and 60 and simultaneously have increased duration between age 60 and 70. Some might have persistent normal or increased durations between age 50 and 60 and simultaneously have increased durations between age 60 and 70. If these populations are not distinguished in Cox analyses, the related results are likely to be false positives driven by the short sleep durations between age 50 and 60.

Besides, the authors did not state why they conducted analyses in two age-scale but not in one.

Similarly, the authors cannot preclude the influence of duration at age 50 on durations at age 60 or age 70.

5. Please state the method used for assessment of the correlation between questionnaire and accelerometer-assessed measures of sleep duration in Statistical analysis.

6. The authors used questionnaires by asking "how many hours of sleep do you have on an average week-night?" to estimate the sleep duration at different age-scale. If collected information reflected recent one year or more sleep duration, the authors should state this point explicitly when introducing the questionnaires because which may bias the associations.

7. When assessing associations of tertiles of objectively assessed sleep duration with the risk of incident dementia, the authors should describe more about the population in accelerometer sub-study, such as how many waves they visit and whether the objectively assessed sleep duration is only collected at baseline.

Furthermore, analyses on accelerometer-assessed sleep duration were not conducted by authors as those seen on questionnaire-assessed duration stratified by age.

8. As well-known, Alzheimer's disease and vascular dementia are common causes of dementia, systematic analyses in multiple types of dementias may offer more interesting findings if the data allow doing.

9. There are occasional grammatical mistakes that need to be corrected.

Reviewer #3 (Remarks to the Author):

This study leverages a unique cohort to examine how sleep duration at different points in midlife

relate to the risk of future dementia. Changes in sleep duration are also examined. The analytical approach is thoughtful and appropriate and the supplemental tables and analyses help to bolster the main findings. The main strengths of the study include the large sample size, the repeated assessments of sleep at different ages, and the use of objective sleep assessments in a subset. Limitations of the study include the methods of dementia case ascertainment (which lack sensitivity/precision), the lack of subtyping for dementia (eg no classification of AD or VaD), and the small number of participants and cases with long sleep duration (although the sample here is still larger than many previous studies).

There is already an extensive literature on sleep duration and dementia. The main contribution of this study is that it examines sleep duration at multiple points in a persons life. This is important since different risk factors can have different effects at different ages (eg with BP being more important at younger ages, including in this cohort). There are already a few studies with substantial follow-up periods (eg Kuopio Ischemic Heart Disease Study with >20 years; Framingham Heart Study with 10 years) and I do not agree that the long follow-up duration used here removes the effect of preclinical dementia (as explained below). However, the data presented here is interesting and informative to know and the analysis is sound. I offer the following comments, mostly around interpretation of data.

Page 8 states that "... short sleep duration is likely to be a risk factor for dementia." Similar things are noted in the last sentence of the abstract. This is speculative and goes beyond the data. As noted in the limitations section, causality cannot be inferred and one cannot rule out residual confounding.

The long follow-up duration is certainly advantageous but does not remove the effect of preclinical dementia entirely. AD can have a preclinical phase lasting 2-3 decades so half your sample that developed dementia may have had some level of AD pathology at the time sleep was first measured.

In some cases, long sleep duration was associated with a higher risk of dementia with a similar effect size to short sleep duration. However long sleep duration was less common leading to a lack of precision around the estimate. Thus, although this sample was very large, it may have been insufficient to completely understand the relationship between sleep time and dementia (particularly at the extreme ends of the sleep time distribution). For example, the Framingham Heart Study reported long sleep (>9 hours) as a risk factor for dementia, but few subjects met this criteria in the current study. Thus, the conclusions reported here may partly stem from the fact that short sleep was more common (and therefore better powered in analysis; a more symmetrical U shape association may exist in the population).

In multiple places it is stated that changes in sleep patterns may be a marker of preclinical dementia or associated with dementia pathophysiology (eg, abstract and conclusion). However, as you know, dementia is a syndrome and not a disease. Dementia has many causes. So, what aspect/type of dementia pathophysiology would drive changes in sleep? Do you think its something specific to AD, since most cases of dementia have contributions from AD?

Were there any interactions with APOE e4 status? This may help to speak to mechanisms. You may also wish to include adjustment for e4 status in statistical models.

REVIEWER 1 COMMENTS

1. I enjoyed reading the ms.

OUR RESPONSE: Thank you for this positive feedback.

2. Table 2: In the age 70 wave, you had 9% fewer subjects compared to the age 60 wave and 18% fewer subjects compared to the age 50 wave, respectively. With this in mind, I would mention in the ms that the lack of significance for the association between self-reported sleep duration and dementia might have been a power issue.

OUR RESPONSE: Thank you, this comment has allowed us to clarify the study design in Supplementary Figure 1 added to the revised manuscript. Not all participants in the analysis of sleep duration at age 50 are included in the analysis of sleep duration at age 70, one reason being participants not reaching the target age by the end of the follow-up. We have added the following text to address the power issue in the discussion section (page 9, paragraph 2):

“As in most previous studies,^{7, 11, 12, 13} the number of long sleepers in our study was small and did not allow a robust estimation of the association with long sleep duration. Two large-scale studies based on individuals with a mean age of 72 years, followed for 12 and 14 years reported sleep duration above 9 hours to be associated with increased risk of dementia.^{14, 15} Given the age of these individuals at the start of follow-up, more data on long sleepers in midlife and old age are needed to draw conclusions on the importance of long sleep duration for dementia.”

We would also like to add that our study remains one of the largest to examine the association of repeat measures of sleep duration and dementia, as can be seen in table 1 on the next page. Most studies have fewer cases than our study (as also acknowledged by reviewer 3 comment 1) and two of the larger studies assessed sleep only at older ages. We have revised the introduction to highlight the importance of the present study in this context (page 3, paragraph 2):

“Much of the evidence on the association between sleep duration and dementia comes from studies with a follow-up of less than 10 years. As most dementias are characterized by pathophysiological changes over 20 years or more,^{9, 10} studies with a long follow-up are needed to provide an insight into the association between sleep duration and subsequent dementia. Among studies with a follow-up of 10 years or longer^{7, 11, 12, 13, 14, 15} many are based on participants 65 years and older at baseline,^{7, 13, 14, 15} not allowing the examination of the importance of sleep characteristics earlier in the lifecourse. The number of dementia cases in the short and long sleep groups in these studies is often small,^{7, 11, 12, 13} leading to imprecise associations due to limited statistical power. Whether the patterns of change in sleep duration leading up to old age is meaningfully associated with incidence of dementia is also unclear.”

Table 1. Summary of studies population characteristics in previous studies on sleep duration and dementia risk with follow-up duration of 10 years or more

Paper	Population study	Mean age at sleep measure	Mean follow-up	N dementia cases (total, and by sleep duration categories)	N total (total, and by sleep duration categories when available)
Bokenberger et al, 2017	Screening across the Lifespan twin study	72 years	14 years	Total: 1844 ≤6 h: 52 7 - 8 h: 1255 ≥9 h: 537	Total: 11247 ≤6 h: not shown 7 - 8 h: not shown ≥9 h: not shown
Larsson et al, 2018	National research infrastructure in Sweden	72 years	13 years	Total: 3671 ≤6 h: 988 6.1 - 7 h: 1110 7.1 - 9 h: 1454 >9 h: 119	Total: 28775 ≤6 h: not shown 6.1 - 7 h: not shown 7.1 - 9 h: not shown >9 h: not shown
Westwood et al, 2018	Framingham heart study	72 years	10 years	Total: 234 <6 h: 24 6 - 9 h: 191 >9 h: 19	Total: 2457 <6 h: 209 6 - 9 h: 2152 >9 h: 96
Lutsey et al, 2018	ARIC study	63 years	15 years	Total: 142 <7 h: 35 7 - <8 h: 50 8 - <9 h: 44 >9 h: 13	Total: 1653 <7 h: 395 7 - <8 h: 627 8 - <9 h: 521 >9 h: 110
Virta et al, 2013	Finnish twin cohort	52 years	23 years	Total: 179 <7 h: not shown 7-8 h: not shown >8 h: not shown	Total: 1144 <7 h: not shown 7-8 h: not shown >8 h: not shown
Luojus et al, 2013	Kuopio Ischemic Heart Disease Study (men)	53 years	22 years	Total: 287 ≤6.5 h: not shown 7-8 h: not shown ≥8.5 h: not shown	Total: 2386 ≤6.5 h: not shown 7-8 h: not shown ≥8.5 h: not shown
OUR PAPER	Whitehall II cohort study	50 years	25 years	Total: 521 ≤6 h: 211 7 h: 219 ≥8 h: 91	Total: 7959 ≤6 h: 3149 7 h: 3624 ≥8 h: 1186
		60 years	15 years	Total: 409 ≤6 h: 192 7 h: 142 ≥8 h: 75	Total: 7164 ≤6 h: 2759 7 h: 2988 ≥8 h: 1417
		70 years	8 years	Total: 392 ≤6 h: 171 7 h: 131 ≥8 h: 90	Total: 6516 ≤6 h: 2429 7 h: 2578 ≥8 h: 1509

References

- Bokenberger K, et al. Association Between Sleep Characteristics and Incident Dementia Accounting for Baseline Cognitive Status: A Prospective Population-Based Study. *J Gerontol A Biol Sci Med Sci* **72**, 134-139 (2017).
- Larsson SC, Wolk A. The Role of Lifestyle Factors and Sleep Duration for Late-Onset Dementia: A Cohort Study. *J Alzheimers Dis* **66**, 579-586 (2018).

- Westwood AJ, *et al.* Prolonged sleep duration as a marker of early neurodegeneration predicting incident dementia. *Neurology* **88**, 1172-1179 (2017).
- Lutsey PL, *et al.* Sleep characteristics and risk of dementia and Alzheimer's disease: The Atherosclerosis Risk in Communities Study. *Alzheimers Dement* **14**, 157-166 (2018).
- Virta JJ, *et al.* Midlife sleep characteristics associated with late life cognitive function. *Sleep* **36**, 1533-1541, 1541A (2013).
- Luojus MK, Lehto SM, Tolmunen T, Brem AK, Lonnroos E, Kauhanen J. Self-reported sleep disturbance and incidence of dementia in ageing men. *J Epidemiol Community Health* **71**, 329-335 (2017).

3. Others have shown that sleep disturbances, such as difficulty initiating or maintaining sleep and sleep apnea are associated with an increased risk for all-cause dementia/AD (PMID 25438949; 21828324). These conditions can result in shorter sleep duration and may account for the observed association between sleep duration and dementia. It would help if you mentioned this.

OUR RESPONSE: We agree; the revised manuscript describes potential mechanisms, including sleep disturbances and apnea, in the discussion. It is likely that sleep apnea is not a major explanatory factor as our estimates of the association between sleep duration and dementia were unchanged after adjustment for obesity which is common in those with sleep apnea. The following paragraph has been added to the revised manuscript (paragraph overlapping pages 8 & 9):

“There is evidence of a bidirectional association between sleep dysfunction and pathophysiological changes in dementia,^{22, 23} highlighting the need for a longer time frame in studies. The results for short sleep duration seen in our study is likely to involve several processes,¹ including neuroinflammation,²⁴ atherosclerosis,²⁵ alpha-synucleinopathies (dementia with Lewy bodies and Parkinson disease dementia),²⁶ and impaired amyloid- β clearance,²⁷ possibly due to impaired glymphatic function.²⁸ Experimental studies support a detrimental effect of sleep deprivation on cognitive performance²⁹ and β -amyloid (A β) clearance.^{30, 31, 32} Amyloid plaque build-up contributes to poor sleep in older adults through its direct impact on sleep-wake regulator brain regions.^{22, 23} There is also some evidence of an association of A β accumulation with disruption of the circadian rhythm and sleep pattern in cognitively normal adults.³³ Much of the research on mechanisms focusses on sleep disturbance rather than sleep duration, particularly in relation to the years leading up to dementia diagnosis. The mechanisms linking short sleep duration to dementia may be similar^{1, 34} but a better understanding of how sleep features (duration, disturbance, sleep apnea, and sleep-wake regulation) over the adult lifecourse shape risk of dementia at older ages is required to identify windows of opportunity for therapeutic interventions to reduce the risk or delay the progression of dementia and its subtypes.”

4. If I understood your analysis correctly, you have, for instance, investigated the association of sleep duration reported at the age 50 wave with dementia risk during the entire observational period. For me, it is surprising that you did not consider sleep duration as a time-varying exposure in your main analysis? I am pretty sure that you will still see the association between sleep duration and dementia if you run a time-updated cox regression. It is a strength to use time-varying data for exposure and covariates in your analysis!

OUR RESPONSE: Thank you, we agree that the study design is complex and we have added a flow chart (Supplementary Figure 1) to reflect the objective of using this study design. Most cases of dementia onset in our study (88%), like in other studies, is after the age of 70 years. By fixing the age at exposure (sleep duration at age 50, 60, and 70) our aim is to capture changes in the nature of the

association between sleep duration and dementia. Sleep duration at 70 years, compared to sleep duration at 50 years, is more likely to be affected by preclinical dementia.

Use of the time-varying approach would involve splitting the time in study period in 3 (50 to 59 years, 60 to 69 years, and 70 years and above) to study association between sleep duration and dementia within each period. As most cases of dementia in our study are after 70 years (88%), this approach boils down to studying the association between sleep duration at age 70 and subsequent risk of dementia. This is not the research question addressed in our study, leading us not to use the time varying approach. Our focus is on whether sleep duration over the adult lifecourse is associated with dementia, with the explicit assumption that when sleep duration is measured late in life (i.e. close to dementia onset) it is likely to reflect bidirectional effects as dementia is known to be characterized by a long preclinical period.

5. Regarding the sleep duration stability analysis: Do you see any differences in the dementia risk between those who have stable sleep duration across all waves (i.e., 50, 60, and 70) and those who do not have stable sleep duration?

OUR RESPONSE: We thank the reviewer for this comment. In the revised manuscript we have added analysis that examines the association of stability of sleep duration between age 50 and 70 with incident dementia. To do this, we first characterised trajectories of sleep duration between age 50 and 70 and then the association of these trajectories (persistent short sleep, persistent normal sleep, persistent long sleep, change from short to normal sleep, change from normal to long sleep, and change from normal to short sleep) with subsequent dementia was examined. These analyses are reported in the revised methods and results sections as follows:

Methods section (page 11, last paragraph)

“Change in sleep duration between age 50 and 70 was defined using group-based trajectory modelling, fitted using the traj-command in Stata.⁴³ The trajectories were chosen based on model fit statistics (Bayesian Information Criterion values and average posterior probabilities) and judgement about whether they adequately addressed the research question.⁴⁴ The sleep duration categories in the construction of trajectories was the same as in the main analyses, and was based on participants who were alive and free of dementia at age 70 with at least 2 out of 3 measures of sleep duration at age 50, 60, and 70.”

Results section (page 5, paragraph 2)

“A total of 6875 participants were alive, free of dementia at age 70, and had at least 2 out of the 3 measures of sleep duration at age 50, 60, and 70. Using these data on sleep duration, six trajectories were identified and labelled as: Persistent short sleep, Persistent normal sleep, Persistent long sleep, Change from short to normal sleep, Change from normal to long sleep, and Change from normal to short sleep (Figure 1). Persistent short sleep duration was associated with an increased risk of dementia (HR=1.30, 95%CI=1.00 to 1.69) compared to those with persistent normal sleep duration (Table 3). There was also a signal of higher risk in participants with persistent long sleep and those who reported short sleep at least once but the associations did not reach statistical significance.”

6. Some experimental studies have investigated the effects of short periods of sleep loss on brain health. It would help if you discussed these study findings, as they are in line with your findings that insufficient sleep duration harms the brain. (PMID 31915189; 29632177; 24887018).

OUR RESPONSE: We thank the reviewer for these references which have been included in the

revised manuscript. We have extended our paragraph on potential mechanisms linking sleep to dementia risk and reported results from experimental studies showing short term effect of sleep deprivation on Alzheimer's disease biomarkers as follows (page 9, first paragraph):

"Experimental studies support a detrimental effect of sleep deprivation on cognitive performance²⁹ and β -amyloid (A β) clearance.^{30, 31, 32"}

7. Given the observed associations between sleep duration and all-cause dementia, I wonder whether you see similar sleep duration and dementia subtypes associations. In this context, AD may be particularly interesting, as accumulating evidence suggests that sleep disruptions (including short sleep duration) increase the risk of AD (e.g., see 26996255).

OUR RESPONSE: This is a good point, although the sub-types of dementia are not comprehensively documented in the electronic health records we used to ascertain dementia. In order to address this issue we undertook sensitivity analysis using a proxy for Alzheimer's disease dementia which was dementia cases without history of cardiovascular disease over the follow-up. A total of 77.5% (N=404) of dementia cases in our study were in this category, which is similar to that reported in the literature (e.g. 2020 Alzheimer's Association report).

In the Methods section, the analysis is presented as follows (page 15, first paragraph):

"Three, in order to examine whether observed results apply to dementia due Alzheimer's disease, we repeated the analyses using a proxy definition of this outcome. This consisted of defining Alzheimer's disease dementia as dementia cases free of cardiovascular disease (stroke or myocardial infarction)⁵³ over the follow-up. Analyses 2 and 3 were exploratory and due to smaller numbers in analysis the direction of results rather than statistical significance should be given more importance."

Results are presented as follows (page 6, last paragraph):

"Use of dementia without history of cardiovascular disease as a proxy for Alzheimer's disease type of dementia (N cases=404, 77.5% of the 521 all cause dementia cases) yielded results similar to those in the main analysis although the estimates were imprecise due to smaller numbers in analysis (Supplementary Table 5)"

In the limitations section, we highlight the lack of comprehensive information on dementia type in electronic health records (page 10, paragraph 1):

"Data on dementia type were incomplete in the records and our use of a proxy measure of Alzheimer's disease dementia may not be precise despite the proportion of cases being in accordance with that in the general population.^{10"}

Reference:

- Alzheimer's Association. 2020 Alzheimer's disease facts and figures. *Alzheimers Dement* **16**, 391-460 (2020).

8. Table 1: Please expand this table by showing the descriptive for the sleep duration groups at age 60 and 70.

OUR RESPONSE: Thank you for the suggestion, we have now added Supplementary Table 1 that shows sample characteristics' by sleep duration groups at age 60 and 70.

REVIEWER 2 COMMENTS

1. The authors studied a considerable population with the long-term monitor on sleep duration and dementia development and constituted three important findings despite they only concentrated on sleep duration but no other sleep characteristics (such as sleep disorders and sleep quality). The analyses from this cohort strengthened the role of sleep hygiene as one modifiable factor in understanding mechanisms and establishments of preventable strategies of incipient dementias. But some major questions are required to be addressed by the authors before considering accepting.
OUR RESPONSE: Thank you. We have revised the paper to include the reviewer's suggestions.

2. Although only three aspects resulted in censored data, listing max and min follow-up duration in total participants and those at different age-scale groups as well as numbers of cases and controls in different follow-up durations would be good to assess analytic rationality.

OUR RESPONSE: As suggested, we have added the range of follow-up for each analysis in the tables. We have also added cumulative hazards of dementia plots for the analyses with sleep duration at each age of interest which provides information on the distribution of dementia cases over the follow-up. This information has been added to the Results section (page 4, first paragraph):

“Cumulative hazards of dementia as a function of sleep duration at age 50, 60, and 70 are presented in Supplementary Figure 2, and show that most cases of dementia were diagnosed after the age of 70 years. Mean age at diagnosis was 77.1 (SD=5.6; range=53.4 to 87.6) years.”

3. “Participants with normal sleep duration more likely to have a better cardiovascular and mental health profile” (in the Results), authors needed to make the definitions of cardiovascular and mental health profile in the Methods because they only mentioned categories of health-related factors. Besides, whether p values were derived from multiple comparisons of group differences in short, normal, and long sleep durations in Table 1 was not stated.

OUR RESPONSE: Thank you for this helpful comment. We have now added the following text to the methods section to describe cardiometabolic and mental health measures (page 13, paragraph 2):

“Health-related variables included measures of cardiometabolic risk factors and mental health. Cardiometabolic factors were hypertension (systolic/diastolic $\geq 140/90$ mmHg or use of antihypertensive medication), diabetes mellitus (determined by fasting glucose ≥ 7.0 mmol/l, reported doctor-diagnosed diabetes, use of diabetes medication, or hospital record), body-mass index (BMI, categorized as <20 , $20-24.9$, $25-29.9$, and ≥ 30 kg/m²) based on height and weight assessment at the clinical examination using standard clinical protocols, and cardiovascular disease (including coronary heart disease and stroke identified using linkage to national hospital records). Mental health factors included current depressive symptomatology defined by the four-item depression subscale of the General Health Questionnaire,⁴⁸ and self-reported use of CNS medication (anti-depressant, antipsychotic, hypnotic, anxiolytic, or Parkinson medications).”

We have added information on P-values in Table 1 in footnotes as follows:

“P for heterogeneity were estimated using χ^2 test for categorical variables, and one-way ANOVA for continuous variables by sleep duration groups and student's t test by dementia groups for continuous variables.”

4. How to understand the observations shown to be a non-significant association of ≤ 5 hours sleep duration in participants at age 50 but significant associations in those at age 60 and 70?

What's more, the authors found the decrease or increase in sleep duration between age 60 and 70 were both associated with risk of dementia when merged subgroups, while in e-Table 2, significant estimates were not noted for most increased or decreased durations. I suggest the authors discuss these points.

OUR RESPONSE: Thank you, we agree with the reviewer. These additional analyses which were included in the supplementary tables were based on a small number of cases (less than 10 dementia cases in some subgroups) are not useful. Our intention was to use finer categories of sleep duration than those used in the main analyses. It is worth pointing out that estimates were similar using detailed categories of sleep duration (5 classes) and wider (3 classes) categories, although the p values were affected in the analyses with detailed categories due to smaller number of cases. Both approaches yield similar results, albeit with imprecision when analyses are based on small numbers. However, we feel that these detailed analyses may appear to add noise to the results rather than help with their interpretation, leading us to remove them from the revised manuscript.

In order to provide consistent results we have revised the analyses on change in sleep duration. Rather than show change in two blocks (from 50 to 60 and then from 60 to 70 years) we now examining change in one block using data from sleep duration at age 50, 60, and 70. Please see response to the next comment for further details on the new analysis on change in sleep duration.

5. In main analyses, the authors noted that changes in sleep durations between age 60 and 70 were associated with the risk of dementia without stating details for those who had changed durations. Some participants might have persistent short durations between age 50 and 60 and simultaneously have increased duration between age 60 and 70. Some might have persistent normal or increased durations between age 50 and 60 and simultaneously have increased durations between age 60 and 70. If these populations are not distinguished in Cox analyses, the related results are likely to be false positives driven by the short sleep durations between age 50 and 60.

Besides, the authors did not state why they conducted analyses in two age-scale but not in one.

Similarly, the authors cannot preclude the influence of duration at age 50 on durations at age 60 or age 70.

OUR RESPONSE: We are grateful to the reviewer for this excellent suggestion of examining change in sleep duration in one block rather than splitting it between 50 to 60 years and 60 to 70 years. We undertook new analysis using the entire period (see Figure 1 and Table 3), the methods and results sections have been modified as follows:

Methods section (page 11, last paragraph)

“Change in sleep duration between age 50 and 70 was defined using group-based trajectory modelling, fitted using the traj-command in Stata.⁴³ The trajectories were chosen based on model fit statistics (Bayesian Information Criterion values and average posterior probabilities) and judgement about whether they adequately addressed the research question.⁴⁴ The sleep duration categories in the construction of trajectories was the same as in the main analyses, and was based on participants who were alive and free of dementia at age 70 with at least 2 out of 3 measures of sleep duration at age 50, 60, and 70.”

Results section (page 5, paragraph 2)

“A total of 6875 participants were alive, free of dementia at age 70, and had at least 2 out of the 3 measures of sleep duration at age 50, 60, and 70. Using these data on sleep duration, six trajectories were identified and labelled as: Persistent short sleep, Persistent normal sleep, Persistent long sleep, Change from short to normal sleep, Change from normal to long sleep, and Change from normal to short sleep (Figure 1). Persistent short sleep duration was associated with an increased risk of dementia (HR=1.30, 95%CI=1.00 to 1.69) compared to those with persistent normal sleep duration (Table 3). There was also a signal of higher risk in participants with persistent long sleep and those who reported short sleep at least once but the associations did not reach statistical significance.”

6. Please state the method used for assessment of the correlation between questionnaire and accelerometer-assessed measures of sleep duration in Statistical analysis.

OUR RESPONSE: We have now added this information as requested (page 5, last paragraph):

“The Pearson’s correlation between questionnaire and accelerometer-assessed measures of sleep duration in 2012-2013 was 0.41 ($p < 0.001$).”

7. The authors used questionnaires by asking “how many hours of sleep do you have on an average week-night?” to estimate the sleep duration at different age-scale. If collected information reflected recent one year or more sleep duration, the authors should state this point explicitly when introducing the questionnaires because which may bias the associations.

OUR RESPONSE: The question was asked exactly as it is described in the manuscript, without reference to time or age. This question has been collected at each data wave in the study and in order to extract sleep duration at age 50, 60, and 70 for each participant we combed through multiple waves of data (as shown in Supplementary Figure 1 in the revised manuscript).

8. When assessing associations of tertiles of objectively assessed sleep duration with the risk of incident dementia, the authors should describe more about the population in accelerometer sub-study, such as how many waves they visit and whether the objectively assessed sleep duration is only collected at baseline.

Furthermore, analyses on accelerometer-assessed sleep duration were not conducted by authors as those seen on questionnaire-assessed duration stratified by age.

OUR RESPONSE: Thank you for these suggestions. We added Supplementary Table 2 to present characteristics of the study population in 2012-2013 by tertiles of accelerometer-assessed sleep duration. The accelerometer sub-study was conducted only once, as an add-on to data collection wave in 2012-2013. Changes appear in the Methods and Results section as follows:

Methods section (page 12, first paragraph) :

“The accelerometer substudy was undertaken in 2012-2013²⁰ on study participants who either attended the central London research clinic or were seen at home in those living in the South-Eastern regions of England. [...] Study participants were 60 and 83 years in this sub-study, a one-off addition to the main data collection. Therefore, age-specific analysis as in the self-reported measure of sleep duration was not possible with the accelerometer data.”

Results section (page 5, last paragraph):

“Data on accelerometer-assessed sleep duration collected in 2012-2013 were available on 3888 participants, among whom 111 developed dementia over a mean 6.4-year follow-up period. Characteristics of this analytical sample are presented in Supplementary Table 2.”

9. As well-known, Alzheimer's disease and vascular dementia are common causes of dementia, systematic analyses in multiple types of dementias may offer more interesting findings if the data allow doing.

OUR RESPONSE: This is a good point, also raised by Reviewer 1, comment 7. Given that causes of dementia were not documented in a comprehensive manner in the electronic health records, we have now added sensitivity analysis for a proxy definition of Alzheimer's dementia by considering dementia cases without CVD history over the follow-up. A total of 77.5% (N=404) of dementia cases met these criteria, which is similar to that reported in the literature (e.g. 2020 Alzheimer's Association report). We did not examine the association of sleep duration with dementia of vascular type to avoid the analyses to be based on very small numbers of cases in some groups of sleep duration that could lead to spurious findings.

In the Methods section, the analysis is presented as follows (page 15, paragraph 3):

"Three, in order to examine whether observed results apply to dementia due Alzheimer's disease, we repeated the analyses using a proxy definition of this outcome. This consisted of defining Alzheimer's disease dementia as dementia cases free of cardiovascular disease (stroke or myocardial infarction)⁵³ over the follow-up. Analyses 2 and 3 were exploratory and due to smaller numbers in analysis the direction of results rather than statistical significance should be given more importance."

Results are presented as follows (page 6, last paragraph):

"Use of dementia without history of cardiovascular disease as a proxy for Alzheimer's disease type of dementia (N cases=404, 77.5% of the 521 all cause dementia cases) yielded results similar to those in the main analysis although the estimates were imprecise due to smaller numbers in analysis (Supplementary Table 5)"

In the limitations section, we highlight the lack of comprehensive information on dementia type in electronic health records (page 10, paragraph 1):

"Data on dementia type were incomplete in the records and our use of a proxy measure of Alzheimer's disease dementia may not be precise despite the proportion of cases being in accordance with that in the general population.^{10"}

Reference:

- Alzheimer's Association. 2020 Alzheimer's disease facts and figures. *Alzheimers Dement* **16**, 391-460 (2020).

10. There are occasional grammatical mistakes that need to be corrected.

OUR RESPONSE: The manuscript has been revised carefully.

REVIEWER 3 COMMENTS

1. This study leverages a unique cohort to examine how sleep duration at different points in midlife relate to the risk of future dementia. Changes in sleep duration are also examined. The analytical approach is thoughtful and appropriate and the supplemental tables and analyses help to bolster the main findings. The main strengths of the study include the large sample size, the repeated

assessments of sleep at different ages, and the use of objective sleep assessments in a subset. Limitations of the study include the methods of dementia case ascertainment (which lack sensitivity/precision), the lack of subtyping for dementia (eg no classification of AD or VaD), and the small number of participants and cases with long sleep duration (although the sample here is still larger than many previous studies).

OUR RESPONSE: We thank the reviewer for this summary of the paper. We have revised the manuscript to address the reviewer's comments, as described below.

2. There is already an extensive literature on sleep duration and dementia. The main contribution of this study is that it examines sleep duration at multiple points in a persons life. This is important since different risk factors can have different effects at different ages (eg with BP being more important at younger ages, including in this cohort). There are already a few studies with substantial follow-up periods (eg Kuopio Ischemic Heart Disease Study with >20 years; Framingham Heart Study with 10 years) and I do not agree that the long follow-up duration used here removes the effect of preclinical dementia (as explained below). However, the data presented here is interesting and informative to know and the analysis is sound. I offer the following comments, mostly around interpretation of data.

OUR RESPONSE: We thank the reviewer for these comments. We agree that observation data are problematic for causal interpretation and we have revised the manuscript accordingly. Please see our response to other comments below.

3. Page 8 states that "... short sleep duration is likely to be a risk factor for dementia." Similar things are noted in the last sentence of the abstract. This is speculative and goes beyond the data. As noted in the limitations section, causality cannot be inferred and one cannot rule out residual confounding.

OUR RESPONSE: We agree. We have revised the manuscript in several places to avoid statements on causation. For example, the sentence mentioned by the reviewer on page 8 (2nd paragraph) has been replaced with the following sentence:

"Results in our study were robust to adjustments for a wide range of covariates, including cardiovascular disease,¹⁶ suggesting a consistent association between short sleep duration in midlife and incident dementia."

4. The long follow-up duration is certainly advantageous but does not remove the effect of preclinical dementia entirely. AD can have a preclinical phase lasting 2-3 decades so half your sample that developed dementia may have had some level of AD pathology at the time sleep was first measured.

OUR RESPONSE: We agree with the reviewer that the preclinical phase of dementia is long although the precise length of this phase is unknown at the present time. As the reviewer knows, much of the research on risk factors for dementia is based on studies where participants are 65 years and older at baseline and the follow-up is less than 10 years. Within this context, it is an advantage to have measures of sleep duration at age 50, 60, and 70 years and a long follow-up. Given uncertainties about the length of the preclinical phase, we have revised the introduction, interpretation of our findings, and the discussion. Below are some examples of the revised text.

Introduction (page 3, paragraph 2)

"Much of the evidence on the association between sleep duration and dementia comes from studies with a follow-up of less than 10 years. As most dementias are characterized by pathophysiological changes over 20 years or more,^{9,10} studies with a long follow-up are

needed to provide an insight into the association between sleep duration and subsequent dementia. Among studies with a follow-up of 10 years or longer^{7, 11, 12, 13, 14, 15} many are based on participants 65 years and older at baseline,^{7, 13, 14, 15} not allowing the examination of the importance of sleep characteristics earlier in the lifecourse. The number of dementia cases in the short and long sleep groups in these studies is often small,^{7, 11, 12, 13} leading to imprecise associations due to limited statistical power. Whether the patterns of change in sleep duration leading up to old age is meaningfully associated with incidence of dementia is also unclear.”

Discussion (paragraph overlapping pages 8 & 9)

“There is evidence of a bidirectional association between sleep dysfunction and pathophysiological changes in dementia,^{22, 23} highlighting the need for a longer time frame in studies. The results for short sleep duration seen in our study is likely to involve several processes,¹ including neuroinflammation,²⁴ atherosclerosis,²⁵ alpha-synucleinopathies (dementia with Lewy bodies and Parkinson disease dementia),²⁶ and impaired amyloid- β clearance,²⁷ possibly due to impaired glymphatic function.²⁸ Experimental studies support a detrimental effect of sleep deprivation on cognitive performance²⁹ and β -amyloid (A β) clearance.^{30, 31, 32} Amyloid plaque build-up contributes to poor sleep in older adults through its direct impact on sleep-wake regulator brain regions.^{22, 23} There is also some evidence of an association of A β accumulation with disruption of the circadian rhythm and sleep pattern in cognitively normal adults.³³ Much of the research on mechanisms focusses on sleep disturbance rather than sleep duration, particularly in relation to the years leading up to dementia diagnosis. The mechanisms linking short sleep duration to dementia may be similar^{1, 34} but a better understanding of how sleep features (duration, disturbance, sleep apnea, and sleep-wake regulation) over the adult lifecourse shape risk of dementia at older ages is required to identify windows of opportunity for therapeutic interventions to reduce the risk or delay the progression of dementia and its subtypes.”

Conclusion (page 10, last paragraph)

“There is a widely acknowledged association between sleep and cognitive function,¹ primarily due to the role of sleep in learning and memory, synaptic plasticity and waste clearance from the brain.^{1, 38, 39} Whether sleep parameters also affect late-life dementia remains the subject of debate. While incipient dementia is known to affect sleep-wake cycles,^{1, 2, 7} the extent to which sleep duration over the adult lifecourse is associated with late-onset dementia is unclear because most studies have not explicitly considered age at assessment of sleep duration or the length of follow-up. Our approach pays attention to both these aspects along with inclusion of a wide array of covariates to show that short sleep duration in midlife is associated with an increased risk of dementia. Public health messages to encourage good sleep hygiene,⁴⁰ may be particularly important for people at higher risk of dementia.”

5. In some cases, long sleep duration was associated with a higher risk of dementia with a similar effect size to short sleep duration. However long sleep duration was less common leading to a lack of precision around the estimate. Thus, although this sample was very large, it may have been insufficient to completely understand the relationship between sleep time and dementia (particularly at the extreme ends of the sleep time distribution). For example, the Framingham Heart Study reported long sleep (>9 hours) as a risk factor for dementia, but few subjects met this criteria in the current study. Thus, the conclusions reported here may partly stem from the fact that short sleep was more common (and therefore better powered in analysis; a more symmetrical U shape association may exist in the population).

OUR RESPONSE: We agree with the reviewer that our analysis on long sleep might be underpowered. The significant association for long sleep in the Framingham heart study was based on 19 dementia cases in 96 participants; the small numbers make it difficult to draw firm conclusions on the robustness of these findings. Please see below the revised paragraph of the discussion on long sleep duration (page 9, 2nd paragraph):

“In contrast to some previous studies,^{7, 11, 14, 15} we did not find strong evidence to support the hypothesis that long sleep duration is associated with dementia. As in most previous studies,^{7, 11, 12, 13} the number of long sleepers in our study was small and did not allow a robust estimation of the association with long sleep duration. Two large-scale studies based on individuals with a mean age of 72 years, followed for 12 and 14 years reported sleep duration above 9 hours to be associated with increased risk of dementia.^{14, 15} Given the age of these individuals at the start of follow-up, more data on long sleepers in midlife and old age are needed to draw conclusions on the importance of long sleep duration for dementia.”

6. In multiple places it is stated that changes in sleep patterns may be a marker of preclinical dementia or associated with dementia pathophysiology (eg, abstract and conclusion). However, as you know, dementia is a syndrome and not a disease. Dementia has many causes. So, what aspect/type of dementia pathophysiology would drive changes in sleep? Do you think its something specific to AD, since most cases of dementia have contributions from AD?

OUR RESPONSE: Thank you, we agree that there are multiple causes of dementia and that dementia is a syndrome. In order to address the comment on possible mechanisms underlying changes in sleep regulation in the preclinical phase of dementia we have revised the paragraph on potential mechanisms as follows (paragraph overlapping pages 8 & 9):

“There is evidence of a bidirectional association between sleep dysfunction and pathophysiological changes in dementia,^{22, 23} highlighting the need for a longer time frame in studies. The results for short sleep duration seen in our study is likely to involve several processes,¹ including neuroinflammation,²⁴ atherosclerosis,²⁵ alpha-synucleinopathies (dementia with Lewy bodies and Parkinson disease dementia),²⁶ and impaired amyloid- β clearance,²⁷ possibly due to impaired glymphatic function.²⁸ Experimental studies support a detrimental effect of sleep deprivation on cognitive performance²⁹ and β -amyloid (A β) clearance.^{30, 31, 32} Amyloid plaque build-up contributes to poor sleep in older adults through its direct impact on sleep-wake regulator brain regions.^{22, 23} There is also some evidence of an association of A β accumulation with disruption of the circadian rhythm and sleep pattern in cognitively normal adults.³³ Much of the research on mechanisms focusses on sleep disturbance rather than sleep duration, particularly in relation to the years leading up to dementia diagnosis. The mechanisms linking short sleep duration to dementia may be similar^{1, 34} but a better understanding of how sleep features (duration, disturbance, sleep apnea, and sleep-wake regulation) over the adult lifecourse shape risk of dementia at older ages is required to identify windows of opportunity for therapeutic interventions to reduce the risk or delay the progression of dementia and its subtypes.”

7. Were there any interactions with APOE e4 status? This may help to speak to mechanisms. You may also wish to include adjustment for e4 status in statistical models.

OUR RESPONSE: Data on APOE ϵ 4 status were available on a subset of participants and we conducted sensitivity analysis to examine its role. This is described as follows in the revised manuscript.

Methods section (page 15, first paragraph)

“Two, in a subsample of participants with data on APOE genotype, we repeated the main analyses with APOE ϵ 4 (1 or 2, versus 0) as a covariate in the analysis.”

Results section (page 6, last paragraph)

“Adding apolipoprotein E (APOE) ϵ 4 as a covariate in the analysis did not alter observed associations between sleep duration at age 50, 60, and 70 and risk of dementia (Supplementary Table 4).”

REVIEWER COMMENTS

Reviewer #2 (Remarks to the Author):

Most questions have been addressed well by authors. Only small points need to be further discussed and reconsidered.

The authors re-analyzed the change in sleep duration between age 50 and 70 in association with dementia risk, including six trajectories. I wonder whether the authors excluded the participants diagnosed with dementia before age 70 and how they treat the influences of changed sleep duration at age 60. In Figure 1, I found that the authors did not mention some trajectories, such as contrary changes among the three age stages. Maybe authors could consider adding changed sleep duration at age 60 into analyses of sleep duration changes between age 50 and age 70 but still in one age-scale.

Reviewer 2 Comment

Most questions have been addressed well by authors. Only small points need to be further discussed and reconsidered.

The authors re-analyzed the change in sleep duration between age 50 and 70 in association with dementia risk, including six trajectories. I wonder whether the authors excluded the participants diagnosed with dementia before age 70 and how they treat the influences of changed sleep duration at age 60. In Figure 1, I found that the authors did not mention some trajectories, such as contrary changes among the three age stages. Maybe authors could consider adding changed sleep duration at age 60 into analyses of sleep duration changes between age 50 and age 70 but still in one age-scale.

OUR RESPONSE: We thank the reviewer for acknowledging the changes in the manuscript at the first revision.

The reviewer raises the question of how analysis between age 50 and 70 was handled in the revised manuscript. In the original submission, we had examined change between age 50 and 70 in two blocks: between age 50 and 60 and then between age 60 and 70. We agreed with comments from reviewers (revision 1) that it was better to examine trajectories in sleep duration between age 50, 60 and 70 as a whole rather than in two blocks. Reviewer 2's comments above seem to suggest that our description of the analyses was not clear. We would like to clarify the following points:

- 1) Data on sleep duration at age 50, 60, and 70 was used to construct trajectories of sleep duration between age 50 and 70.
- 2) The identification of trajectories uses all the data and the fit statistics are used to choose the number and pattern of trajectories that best fit observed data. Thus, the comment "I found that the authors did not mention some trajectories, such as contrary changes among the three age stages" is explained by the fact that such a group was not seen in the data. The method used categories all participants into one of the six trajectories (Figure 1). Participants cannot be assigned to a trajectory not seen in the data.
- 3) Please note that all analyses were on incident dementia. In the association between trajectories of sleep duration over the age period of 50, 60 and 70 and incident dementia all participants were free of dementia at age 70.

Sections of the revised manuscript related to this analysis are presented below:

Methods section (revised)

Page 11, last paragraph

"Group-based trajectory models with data on sleep duration at age 50, 60, and 70 were fitted using the *traj*-command in Stata.⁴³ The trajectories were chosen based on model fit statistics (Bayesian Information Criterion values and average posterior probabilities) and judgement about whether they adequately addressed the research question.⁴⁴ The sleep duration categories in the construction of trajectories were the same as in the main analyses, and were based on participants who were alive and free of dementia at age 70 with at least 2 out of 3 measures of sleep duration at age 50, 60, and 70."

Page 14, 3rd paragraph

“We examined the association of trajectories of sleep duration between age 50 and 70 (using data from age 50, 60, and 70) with incident dementia with age of entry and covariates drawn from when participants were 70 years.”

Results section (revised)

Page 5, 2nd paragraph

“A total of 6875 participants were alive, free of dementia at age 70, and had at least 2 out of the 3 measures of sleep duration at age 50, 60, and 70. Using these data on sleep duration, participants were assigned to one of the following six trajectory groups labelled as: Persistent short sleep, Persistent normal sleep, Persistent long sleep, Change from short to normal sleep, Change from normal to long sleep, and Change from normal to short sleep (Figure 1). Persistent short sleep duration at age 50, 60, and 70 was associated with an increased risk of incident dementia (HR=1.30, 95%CI=1.00 to 1.69) compared to those with persistent normal sleep duration over the same age period (Table 3).”

Discussion (not revised)

Page 8, 2nd paragraph

“Analysis of trajectories of sleep duration using data from sleep duration at age 50, 60, and 70 showed persistent short sleep duration to be associated with an increased risk of dementia.”

Figure 1 Title (not revised)

“Trajectories of sleep duration using data on sleep duration at 50, 60, and 70 years”

Table 3 Title (not revised)

“Association of trajectories of sleep duration (using data on sleep duration at 50, 60, and 70 years) with incidence of dementia”

REVIEWERS' COMMENTS

Reviewer #2 (Remarks to the Author):

The questions have been well addressed.